# Global land surface 250-m 8-day Fraction of Absorbed Photosynthetically Active Radiation (FAPAR) product from 2000 to 2021

Han Ma[1], Shunlin Liang[1], Changhao Xiong[2], Qian Wang[3], Aolin Jia[4], Bing Li[5]

[1] Department of Geography, the University of Hong Kong, Hong Kong, China
[2] School of Remote Sensing and Information Engineering, Wuhan University, Hubei, 430010, China
[3] Faculty of Geography, Beijing Normal University, Beijing, 100875, China
[4] Department of Geographical Sciences, University of Maryland, College Park, MD, 20742, USA
[5] Key Research Institute of Yellow River Civilization and Sustainable Development & Collaborative Innovation Center on Yellow River Civilization of Henan Province, Henan University, Kaifeng, 475001, China

*Correspondence to*: Shunlin Liang (shunlin@hku.hk)

**Abstract.** The Fraction of Absorbed Photosynthetically Active Radiation (FAPAR) is a critical land surface variable for carbon cycle modeling and ecological monitoring. Several global FAPAR products have been released and have become widely used; however, spatiotemporal inconsistency remains a large issue for the current products, and their spatial resolutions and accuracies can hardly meet the user requirements. An effective solution to improve the spatiotemporal continuity and accuracy of FAPAR products is to take better advantage of the temporal information in the satellite data using deep learning approaches. In this study, the latest version (V6) of the FAPAR product with a 250-m resolution was generated from Moderate Resolution Imaging Spectroradiometer (MODIS) surface reflectance data and other information, as part of the Global Land Surface Satellite (GLASS) products suite. In addition, it was aggregated to multiple coarser resolutions (up to 0.25° and monthly). Three existing global FAPAR products (MODIS Collection 6, GLASS V5, and PROBA-V V1) were used to generate the time series training samples, which were used to develop a Bidirectional Long Short-Term Memory (Bi-LSTM) model. Direct validation using high-resolution FAPAR maps from the Validation of Land European Remote sensing Instrument (VALERI) and ImagineS networks revealed that the GLASS V6 FAPAR product has a higher accuracy than PROBA-V, MODIS, and GLASS V5, with an $R^2$ value of 0.80 and Root Mean Square Errors (RMSEs) of 0.10–0.11 at the 250-m, 500-m, and 3-km scales, and a higher percentage (72%) of retrievals for meeting the accuracy requirement of 0.1. Global spatial evaluation and temporal comparison at the Ameriflux and National Ecological Observatory Network (NEON) sites revealed that the GLASS V6 FAPAR has a greater spatiotemporal continuity and reflects the variations in the vegetation better than the GLASS V5 FAPAR. The higher quality of the GLASS V6 FAPAR is attributed to the ability of the Bi-LSTM model, which involved high-quality training samples and combines the strengths of the existing FAPAR products, as well as the temporal and spectral information from the MODIS surface reflectance data and other information.

## 1 Introduction

Long-term satellite remote sensing observations of terrestrial vegetation are critical to understanding and monitoring climate change. Vegetation influences the global carbon cycle and climate by taking up atmospheric $CO_2$ through photosynthesis (Knorr et al., 2010). This can be constrained by the energy absorption capacity of the vegetation, and the terrestrial variable related to this process is the Fraction of Absorbed Photosynthetically Active Radiation (FAPAR). FAPAR is defined as the fraction of the downward surface visible solar radiation (400–700 nm) absorbed by the green elements of plants (Gower et al., 1999). It has been recognized as one of the essential climate variables (Gcos, 2016) and is directly linked to the productivity of vegetation (Kaminski et al., 2012; Mccallum et al., 2009; Smith et al., 2020). FAPAR is not only dependent on the canopy structure, leaf properties, soil albedo but also dependent on the illumination conditions. FAPAR under direct illumination is called the black-sky FAPAR, and that under diffuse illumination is called the white-sky FAPAR (Baret et al., 2013; Liu et al., 2019).

Monitoring the changes in the FAPAR using satellites sensors plays an important role in energy balance and carbon cycle modeling. Currently, several global moderate-resolution satellite FAPAR products have been generated and released to the public, including the Sea-viewing Wide Field-of-view Sensor (SeaWiFS) (Gobron et al., 2006), Moderate Resolution Imaging Spectroradiometer (MODIS) (Knyazikhin et al., 1998), Medium Resolution Imaging Spectrometer (MERIS) (Bacour et al., 2006), Geoland2/BioPar version 1 (GEOV1) (Baret et al., 2013), PRoject for On-Board Autonomy–Vegetation (PROBA-V) (Fuster et al., 2020), and Global LAnd Surface Satellite (GLASS) (Liang et al., 2021; Xiao et al., 2015) products. These products have been applied in different research areas such as primary productivity calculations (Zhang et al., 2014), the monitoring of vegetation health (Ivits et al., 2016; Cammalleri et al., 2019; Gobron et al., 2005), crop yield investigations (Dong et al., 2016), and carbon cycle data assimilation in terrestrial ecosystems (Knorr et al., 2010; Smith et al., 2020).

There are two typical types of algorithms for retrieving the FAPAR from satellite data: statistical model based and radiative transfer model based algorithms. For example, several studies have estimated FAPAR by establishing the relationship between the Vegetation Index (VI) and in-situ FAPAR measurements using simple statistical methods or machine learning models (Gitelson et al., 2014; Muller et al., 2020; Camacho et al., 2021), while other studies have estimated FAPAR based on energy balance inside the canopy using radiative transfer models (Zhang et al., 2021; Xiao et al., 2015; Liu et al., 2019). Although many satellite products have been generated using these two types of methods, and their accuracies have been evaluated and inter-compared in many studies (Weiss et al., 2014b; Tao et al., 2015; Xiao et al., 2018; Putzenlechner et al., 2019), these algorithms usually only use single-phase remote sensing data, and the critical temporal information contained in the satellite signals is often ignored.

One of the largest problems with the current land surface products is their spatiotemporal inconsistency (Fang et al., 2019; Weiss et al., 2014b). Owing to the presence of clouds and aerosol contamination in the input observation data, the satellite products usually contain data gaps and outliers, and they are also plagued by a serious lack of data under special geographical and meteorological conditions, such as in tropical forests and high-latitude areas (Camacho et al., 2013). Although many studies have tried to fill these data gaps using either statistical temporal and spatial filtering approaches (Weiss et al., 2014a; Li et al., 2017) or data assimilation methods that exploit dynamic models and ancillary variables (Chernetskiy et al., 2017; Ma et al., 2022), their performances are affected by many factors, such as abrupt land surface changes and cloud cover lasting for a long period of time.

In addition, the targeted FAPAR accuracy and spatial resolution required by the Global Climate Observing System (GCOS) are 0.05 and 200 m, respectively. However, the global FAPAR product with the highest spatial resolution (300 m) product, i.e., the PROBA-V product, begins in 2014, which cannot meet the requirement for long time-series data in climate change applications. The reported uncertainties of the current FAPAR products vary from 0.08 to 0.23 (Tao et al., 2015; Weiss et al., 2014b; Pickett-Heaps et al., 2014). Brown et al. (2020) recently evaluated the MODIS, Visible/Infrared Imager/Radiometer Suite (VIIRS), and PROBA-V FAPAR products over North America and revealed that the PROBA-V product has a higher agreement with field reference data and a better temporal continuity than the MODIS FAPAR and VIIRS FAPAR products. There is an urgent need to develop a 250-m FAPAR product from MODIS surface reflectance data owing to the much longer time period of the MODIS data (i.e., from 2000 to present).

The use of the temporal information contained in the original satellite data, which is ignored by the above-mentioned FAPAR estimation algorithms, may be an effective way to improve the spatiotemporal continuity and accuracy of the FAPAR. Recently, by generating a new version of the GLASS LAI product with a 250-m resolution based on a Bidirectional Long Short-Term Memory (Bi-LSTM) deep learning approach (Ma and Liang, 2022), we have demonstrated that the temporal information in satellite observations is extremely useful for generating high-level products with better spatiotemporal continuity and higher accuracy, and the Bi-LSTM model outperforms the general regression neural network (GRNN), LSTM, gated recurrent unit (GRU) in learning the temporal relationship between satellite surface reflectance and vegetation variable. To keep consistency with the GLASS LAI product, in this study, we applied the same strategy to produce a 250-m FAPAR product from MODIS data.

## 2 Data

### 2.1 Satellite products

This deep learning approach for FAPAR production capitalizes on the existing global FAPAR products. Three widely used global FAPAR products (MODIS Collection 6, GLASS V5, and PROBA-V V1 FAPAR) were fused to generate the true values of the FAPAR time series to create a training dataset.

The MODIS 8-day 500-m FAPAR product (MCD15A2H, Collection 6) was inverted from the MODIS red and near-infrared surface reflectance based on look-up tables simulated using a three-dimensional radiative transfer model for eight biome types. When this main algorithm fails, a backup solution that links the Normalized Difference Vegetation Index (NDVI) to FAPAR is adopted (Myneni et al., 2015; Yan et al., 2016). The MODIS FAPAR corresponds to the instantaneous black sky FAPAR values (i.e., under direct illumination) at 10:30 solar time. Since the MODIS FAPAR is not retrieved over barren, permanent snow and ice covered land, the value is set to 0 over these non-vegetated pixels. The Copernicus Global Land Service (CGLS) 10-day 300-m PROBA-V FAPAR (V1) is generated from the PROBA-V blue, red, and Near Infrared (NIR) surface reflectance data using an Artificial Neural Network (ANN) (Baret et al., 2016). The instantaneous FAPAR is estimated first, and then, smoothing and gap filling are applied over a compositing time window. The PROBA-V FAPAR is defined as the instantaneous black sky FAPAR values (i.e., under direct illumination) at 10:00 solar time. The GLASS V5 FAPAR is derived from the GLASS Leaf Area Index (LAI) and clumping index products based on the energy balance inside the canopy and soil (Xiao et al., 2015). It mainly considers transmittance of Photosynthetically Active Radiation (PAR) under direct illumination and represents the clear-sky FAPAR at 10:30 a.m. local time. Although the FAPAR definitions of the three products are somewhat different, according to previous studies, the impact of these differences on FAPAR's largest difference is less than a few percent compared to the uncertainties of the products (Martínez et al., 2013; Weiss et al., 2007). Thus, the differences in the FAPAR definitions were ignored in this study. In addition, these three products can be used to approximate the daily integrated FAPAR, which is more commonly needed by users than the instantaneous FAPAR. This hypothesis is based on several studies that have reported that the instantaneous FAPAR value at 10:00–10:30 (or 14:00–14:30) solar time is very close to the daily average FAPAR value under clear-sky conditions (Baret et al., 2007; Fensholt et al., 2004). Therefore, the goal of this study was to estimate the black-sky FAPAR around 10:30 a.m. solar time, which is an approximation of the daily average FAPAR.

The 8-day GLASS V6 LAI product was adopted as a potential input feature for the model training since the LAI has been recognized as one of the most sensitive parameters for FAPAR estimation in previous studies (Xiao et al., 2015; Liu et al., 2019). The 8-day 500-m and 250-m GLASS LAI (V6) were produced from a Bi-LSTM model and from MODIS surface reflectance data. The time series training samples was generated from three existing LAI products (MODIS, PROBA-V, and GLASS V5) using K-means clustering analysis and the least difference criteria. Direct validation using 79 high-resolution LAI reference maps from three in situ observation networks revealed that the GLASS V6 LAI had the highest accuracy among the

current LAI products, with a Root Mean Square Error (RMSE) of 0.92 at a resolution of 250 m and 0.86 at a resolution of 500

m, while the RMSE of PROBA-V was 0.98 at a resolution of 300 m, and those of the GLASS V5 and MODIS C6 were 1.08

and 0.95, respectively, at a resolution of 500 m (Ma and Liang, 2021).

The MODIS surface reflectance product was used as the observation data for the training of the deep learning model and the

FAPAR estimation. In the model training process, since the time-series of the 8-day FAPAR samples created from the existing

FAPAR products has a 500-m spatial resolution, the 8-day 500-m surface reflectance product (MOD09A1, V6) was used in

the model training process. In addition, the 250-m surface reflectance product (MOD09Q1, V6) aims to produce the 250-m

FAPAR data and only provides the red and NIR bands at 645 and 858 nm. Therefore, only the first two red and NIR bands

(b1, b2) and the three solar and satellite angles (solar zenith angle $\theta_s$, view zenith angle $\theta_v$, and relative azimuth angle $\varphi$) of

MOD09A1 were used in this study.


Although the MODIS surface reflectance has been atmospherically corrected for gases, aerosols, and Rayleigh scattering,

residual noise caused by clouds still exists. To remain consistent with the GLASS LAI algorithm, the surface reflectance was

not smoothed, and only the pixels with reflectance values outside the [0, 1] range, or without atmospheric correction ($\theta_s >$

85°), were set to zero.


Since the MODIS, PROBA-V, and GLASS V5 FAPAR products provide data from 2000, 2014, and 2000 to the present,

respectively, the five overlapping years (2014–2018) were selected to generate the global training samples and to test the

suitable temporal length for producing the FAPAR. To avoid the invalid values contained in the data samples, FAPAR values

outside of the [0, 1] range were set to zero.


## 2.2 Field FAPAR data

The field FAPAR data used to validate the accuracy of the FAPAR products were collected from 38 sites in the Validation of

Land European Remote sensing Instrument (VALERI) (Baret et al., 2005) and ImagineS (Fuster et al., 2020) networks with

different land cover types. The ground data were derived from the Digital Hemispherical Photos (DHP) and represent the

fraction of the intercepted PAR (FIPAR), which is considered to be nearly the same as FAPAR (Brown et al., 2020; Weiss et

al., 2007). The ground FAPAR measurements were locally regressed using the Landsat or SPOT satellite reflectance to

generate 62 high-resolution FAPAR reference maps with scales of 20 or 30 m (Table S1), which were then reprojected onto

the MODIS sinusoidal projection and aggregated to 250-m and 500-m resolutions to validate the FAPAR products at these

resolutions. The high-resolution maps from VALERI and ImagineS were also aggregated to a 3-km resolution, and the other

field FAPAR values that represent a 3-km area were collected from the DRECT dataset (Garrigues et al., 2008). A total of 111

reference values were used to validate the FAPAR products at the 3-km resolution. As the field measured FAPAR is paired

with the field measured LAI, the quality of the field FAPAR was kept consistent with that of the LAI, and the quality was controlled using the relationship between the NDVI and LAI in our previous study (Ma and Liang, 2022).

To evaluate the temporal consistency of the FAPAR products, we collected the time-series field FAPAR data from two Ameriflux sites (Novick et al., 2018): the Bartlett experimental forest site (US-Bar) and the Mead-irrigated maize-soybean rotation site (US-Ne2). The incoming and outgoing flux, and the flux transmitted through the canopy to the ground were sampled at half hour intervals, with tower measurement heights of 25 m and 5 m for the US-Bar and US-Ne2 sites, respectively. The FAPAR was calculated as the ratio of the measured APAR to the PAR, and the sampled FAPAR values were averaged
over the ±30-min time window of the MODIS overpass time (10:30 a.m. local time) to produce field FAPAR references for the product comparison. We also used multi-year field measurements (2014–2020) for 10 National Ecological Observatory Network (NEON) sites (Table S2) for the temporal consistency evaluation of the FAPR products. This dataset was provided by the Ground Based Observations for Validation (GBOV) of the Copernicus Global Land Service (https://land.copernicus.eu/global/gbov/). The field FAPAR of the NEON sites was derived from the DHPs and represent the
instantaneous black-sky FAPAR at 10:00 local time. The spatial representativeness of the NEON sites is about 1.5 km. However, since the Ameriflux and NEON measurements are tower based, their spatial representativeness is not as explicit as that of the VALERI and ImagineS reference data, so they were only used for intercomparison at the 500 m scale.

**3 Methods**

The workflow of the algorithm used to generate the 250-m FAPAR is shown in Fig. 1. A deep learning approach that exploits
the temporal information in the satellite signals and the current products was adopted to produce the 250-m GLASS V6 FAPAR. The MODIS, GLASS V5, and PROBA-V FAPAR products were used to generate the time series of true values of FAPAR for the global representative sample pixels. The Bi-LSTM model, which was used to produce the GLASS V6 LAI, was used to determine the relationship between the time-series FAPAR and the surface reflectance data.

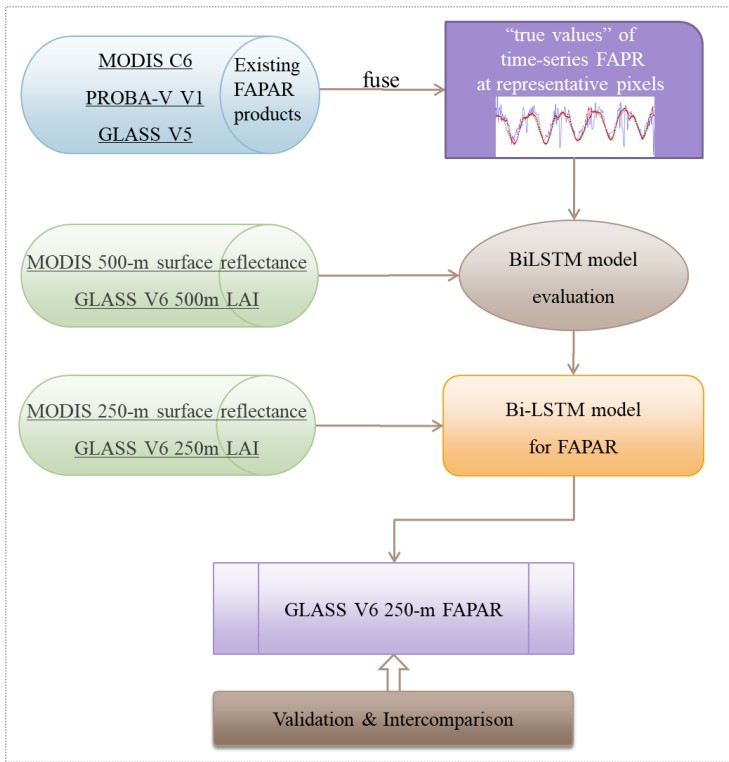

 **Figure 1**: Workflow of the algorithm used to generate the GLASS V6 250-m FAPAR

## 3.1 Creating global training samples

Since FAPAR is physically related to LAI (Mota et al., 2021), the same sample pixels in the GLASS V6 LAI algorithm were used in this study. These globally distributed representative 52,997 sample pixels (Fig. S1) were selected based on global time-series LAI cluster analysis and the least difference criterion and by assuming that the LAI values of the three products with the lowest Mean Square Errors (MSEs) were representative of the true values for a specific pixel (Ma and Liang, 2022).

In each sample pixel that can represent a time-series profile of the land surface and satellite observation conditions, the differences among the MODIS, PROBA-V, and GLASS V5 FAPARs should be the lowest because the LAIs of these products are the least different. To create the time series of the FAPAR true values, for each 8 day time step from 2014 to 2018, if the difference between the PROBA-V and GLASS-V5 FAPARs was less than 0.1, their average was used as the true value; otherwise, the median value of the MODIS, PROBA-V, and GLASS-V5 FAPARs was used. As GLASS V5 and PROBA-V FAPAR have relatively smooth temporal profiles because of their pre- or post- process smoothing algorithms, the created FAPAR samples inherited both their smoothness and accuracy at the selected sample pixels. The histogram distribution of the fused FAPAR values (maximum, average, and all values of the time series) for the representative pixels, and the GLASS V5,

MODIS, and PROBA-V FAPARs for the global land pixels are shown in Fig. 2. Their distributions are quite consistent, indicating that the time-series FAPAR samples are globally representative.

The corresponding 2014–2018 time-series MODIS 500-m surface reflectance (MOD09A1), as well as the GLASS V6 500-m
LAI, for the representative pixels were extracted and were used as the control variables of the data samples. We randomly selected 70% of the samples to train the deep learning model, 20% to select the optimal deep learning model, and 10% to evaluate the final model.

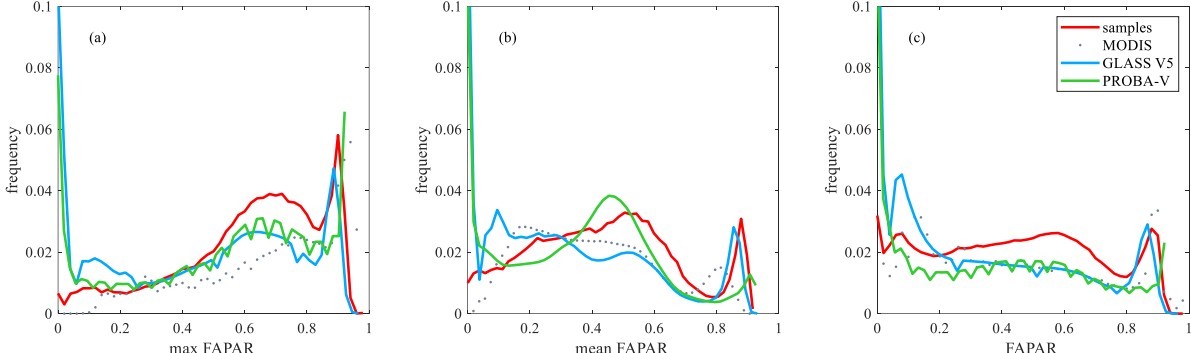

**Figure 2:** Histogram distribution of the FAPAR values (bin width of 0.02) of the 2014–2018 time-series fused FAPAR samples for the representative pixels, and MODIS, PROBA-V, and GLASS V5 FAPARs for the global land pixels: (a) max FAPAR is the distribution of the maximum FAPAR values of the 2014–2018 time series, (b) mean FAPAR is the distribution of the mean FAPAR values of the 2014–2018 time series, and (c) FAPAR is the distribution of all of the FAPAR values of the 2014–2018 time series

### 3.2 Bi-LSTM model

The Bi-LSTM is a variant of the LSTM and contains forward and backward recurrent net layers (Graves and Schmidhuber, 2005), which are connected to each other and to the output layer. Thus, it can process both the previous and future information of the time series during each time step. The LSTM is involved from the Recurrent Neural Network (RNN), which can process a sequence data with gaps using its internal memory state. The LSTM is based on its ability to selectively retain or discard relevant information by modulating the information flow using the input, output, and forget gates and the cell state (Yildirim,
2018). The structure of the Bi-LSTM has been described by (Ma and Liang, 2022).

The structure of the Bi-LSTM model used for the FAPAR estimation had four layers, namely, an input layer, a Bi-LSTM layer with 200 neurons, a dropout layer (5%), and a regression layer. The Adam optimizer, an initial learning rate of 0.0001, a batch size of 100, and a maximum number of epochs of 100 were set as the parameters for the model training. The time series of the red and NIR reflectance, three angles of the MODIS surface reflectance, and the GLASS V6 500-m LAI were set as the features
of the input layer. First, models trained using datasets with two combinations of feature sets (with or without LAI) and a temporal length of 1 year were explored to optimize the input features. Then, the temporal length of the datasets was evaluated to determine the suitable length of the Bi-LSTM model for FAPAR production. Five temporal lengths with a minimum length

of 1 year and a maximum length of 5 years were used in this study. The coefficient of determination ($R^2$), RMSE, and bias were used to evaluate the models' performances. Finally, the Bi-LSTM model with the optimal feature set and temporal length was retrained to obtain the final 250-m FAPAR product model.

## 3.3 Estimating the 250-m resolution FAPAR

The final trained Bi-LSTM model was used to produce FAPAR with an 8-day frequency and 250-m resolution. In line with the model evaluation results (Section 4.1), the MODIS 250-m surface reflectance and the GLASS 250-m LAI were used as the input data of the model, and the optimal temporal length used to estimate the FAPAR was determined to be 3 years. Owing to the training error of the model, the derived time-series FAPAR at the connections of two time windows may be discontinuous, and the same post-processing of the GLASS V6 LAI was adopted here. First, the GLASS FAPAR was calculated in the 2000–2002, 2002–2004,… 2018–2020 time windows, which took about 48 h for a time window using a single Graphic Processing Unit (GPU). Since the connection years (2002, 2004,… 2018) were calculated twice, they were assigned a weight function to obtain the final FAPAR estimates of these years.

The weight function ($w$) of the GLASS V6 LAI algorithm was adopted:

$$
w = \begin{cases}
0 & (1 \leq t \leq 4) \\
0.5 * (cos(\frac{-\pi*t}{37} + \frac{\pi*42}{37}) + 1) & (5 \leq t \leq 42) \\
1 & (43 \leq t \leq 50) \\
0.5 * cos(\frac{\pi*t}{37} - \frac{\pi*51}{37}) + 1)) & (51 \leq t \leq 88) \\
0 & (89 \leq t \leq 92)
\end{cases}
\tag{1}
$$

The FAPAR at time step $t$ for the connection or current year ($fapar_t$) was calculated as follows:

$$
fapar_t = fapar1_{t+46} \cdot w_{t+46} + fapar2_t \cdot w_t \qquad (1 \leq t \leq 46),
\tag{2}
$$

where $fapar1$ is the time-series FAPAR for the previous and current year, and $fapar2$ is that for the current and following year. Taking the year 2002 as an example, $fapar1$ is the time-series FAPAR for 2001–2002, and $fapar2$ is that for 2002–2003. Because MODIS began collecting data on February 24, 2000, for the calculation window of years 2000-2002, the missing data at the beginning days of 2000 was substituted by 2001 to fit the trained model, and the final FAPAR product begins from DOY 57.

## 3.4 Quality assessment of the GLASS V6 FAPAR product

To quantify the accuracy of the GLASS V6 FAPAR product, we extracted its values in the areas of the corresponding VALERI and ImagineS sites at the original 250-m resolution, and we aggregated them to resolutions of 500 m and 3 km by averaging

to enable direct validation at the 250-m, 500-m, and 3-km scales. To compare the three FAPAR products used in this study,
the corresponding MODIS, PROBA-V, and GLASS V5 FAPAR values were also extracted and aggregated.

Then, we assessed the spatial consistency of the GLASS V6 FAPAR product by displaying and analyzing the global
distribution of the FAPAR maps in January and July of 2018, as well as two cloud-dominated and two middle-high-latitude
areas: the Yungui district in Southwestern China, the Congo rainforest in central Africa, the central Europe and Alaska regions.
The temporal consistency of the time-series FAPAR from 2000 to 2020 was also assessed at 18 typical sites with representative
biome types. As last, we demonstrated the 2021 8-day time series FAPAR images at one 1°×1° cloud-dominated region in
Southwestern China for the spatiotemporal consistency assessment.

## 4 Results

### 4.1 Evaluation of Bi-LSTM model

The performances of the Bi-LSTM models trained using datasets with two different combinations of feature sets and a length
of 1 year are shown in Table 1. The results indicate that incorporating the LAI as one feature of the input improves the accuracy
of the model. This finding is consistent with those of many previous studies, that is, the LAI is an important variable for
estimating the FAPAR (Tao et al., 2016).

The evaluation results for different temporal lengths are shown in Fig. 3. The RMSE decreases with increasing temporal length
from 1 to 5 years for the training, validation, and test datasets. However, the turning point is a length of 3 years, so we set 3
years as the suitable temporal length for the production of the FAPAR. Using the optimal features and temporal length, we
retrained the Bi-LSTM model, and the accuracy of the final model is listed in the last row of Table 1.

**Table 1:** Accuracies of Bi-LSTM models with different combinations of features

| | Training | | | Validation | | | Test | | |
|---|---|---|---|---|---|---|---|---|---|
| | $R^2$ | RMSE | Bias | $R^2$ | RMSE | Bias | $R^2$ | RMSE | Bias |
| b1, b2, sza, vza, azi | 0.955 | 0.056 | 0.004 | 0.953 | 0.058 | 0.005 | 0.952 | 0.058 | 0.004 |
| b1, b2, sza, vza, azi, LAI | 0.962 | 0.051 | −0.005 | 0.962 | 0.052 | −0.005 | 0.961 | 0.052 | −0.004 |
| final Bi-LSTM model | 0.964 | 0.050 | 0.000 | 0.964 | 0.050 | 0.000 | 0.963 | 0.051 | 0.000 |

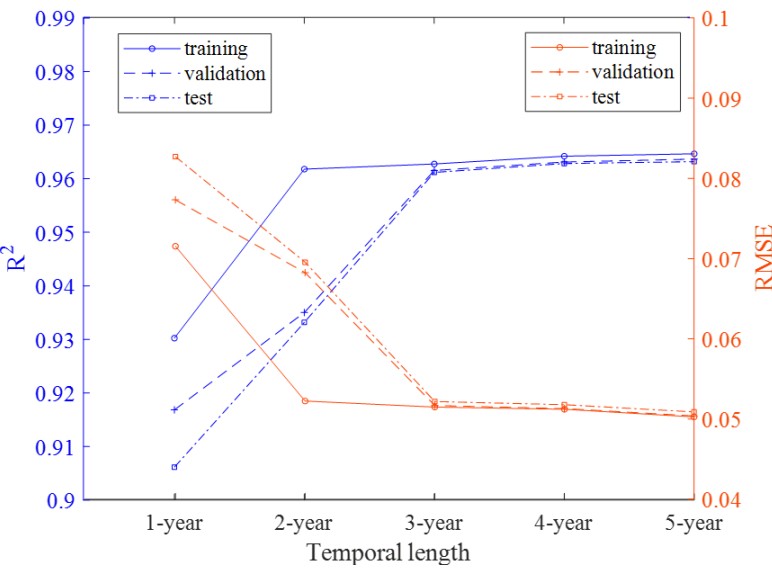

**Figure 3**: Evaluation of temporal length of Bi-LSTM model

## 4.2 Direct validation of FAPAR products

The four FAPAR products were directly validated against field measurements at different scales. The 300-m PROBA-V and
250-m GLASS V6 FAPAR were validated at the 250-m scale, the 500-m MODIS, GLASS V5, and GLASS V6 FAPARs were
validated at the 500-m scale, and all four of the products were aggregated to 3 km and validated at the 3-km scale.

The target accuracy requirement of the GCOS for the global FAPAR product is 0.05; however, meeting this requirement
remains a large challenge due to the uncertainties of both the field measurements and the regression based high-resolution
FAPAR reference maps. Therefore, we used the same FAPAR requirement of 0.1 unit adopted by Brown et al. (2020) to
evaluate the PROBA-V FAPAR product, and denoted a metric P to represent the percentage of pixels meeting the target
accuracy requirement in our validation.

The validation results for the PROBA-V and GLASS V6 FAPAR products at the 250-m scale are shown in Fig. 4. Using 23
upscaled high-resolution FAPAR reference maps for 2014 to 2016 obtained from the ImagineS network, the GLASS V6
achieved a slightly higher accuracy than the PROBA-V, with $R^2$ values of 0.80 and 0.78 and RMSE values of 0.11 and 0.12,
respectively. By adding the remaining 39 reference maps from the VALERI network from 2000 to 2013 (the PROBA-V
FAPAR is not available during this time period), the RMSE of the GLASS V6 was reduced to 0.10 (Fig. 4c).

The validation results for the MODIS, GLASS V5, and GLASS V6 500-m FAPAR products at the 500-m scale based on 62 upscaled reference FAPAR maps are shown in Fig. 5. The GLASS V6 had the highest accuracy ($R^2 = 0.80$, RMSE = 0.10), followed by GLASS V5 and MODIS (both $R^2 = 0.69$, RMSE = 0.13). The MODIS FAPAR had fewer validation points than the GLASS because of the missing data, and it had the largest bias of 0.07 at these sites. The validation results for the MODIS, GLASS V5, and GLASS V6 FAPAR products at the 3-km scale obtained using the DIRECT dataset for 2000 to 2017 are

shown in Fig. 6. The GLASS V6 matched the most data points with the field reference (N=111). Consistent with the 500-m validation results, the GLASS V6 FAPAR had the highest accuracy ($R^2 = 0.81$, RMSE = 0.11).

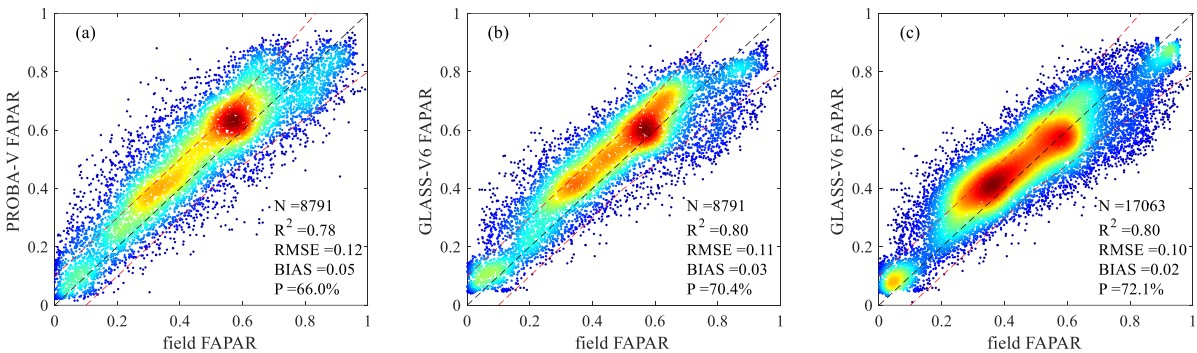

**Figure 4**: Direct validation of the (a) PROBA-V and (b) GLASS V6 FAPAR products at the 250-m scale using 23 upscaled high-resolution FAPAR reference maps from 2014 to 2016 from the ImagineS network; (c) direct validation of the GLASS V6 FAPAR product at the 250-

m scale using 62 high-resolution FAPAR reference maps from 2000 to 2016 from the Bigfoot, VALERI, and ImagineS networks. The red dashed lines denote the accuracy requirement;P

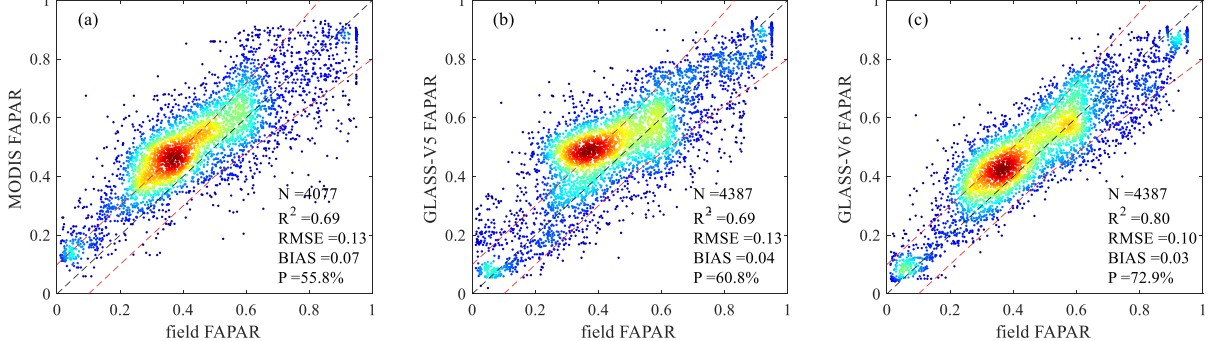

**Figure 5:** Direct validation of the (a) MODIS, (b) GLASS V5, and (c) GLASS V6 FAPAR products at the 500-m scale using 62 upscaled high-resolution FAPAR reference maps from 2000 to 2016 from the Bigfoot, VALERI, and ImagineS networks

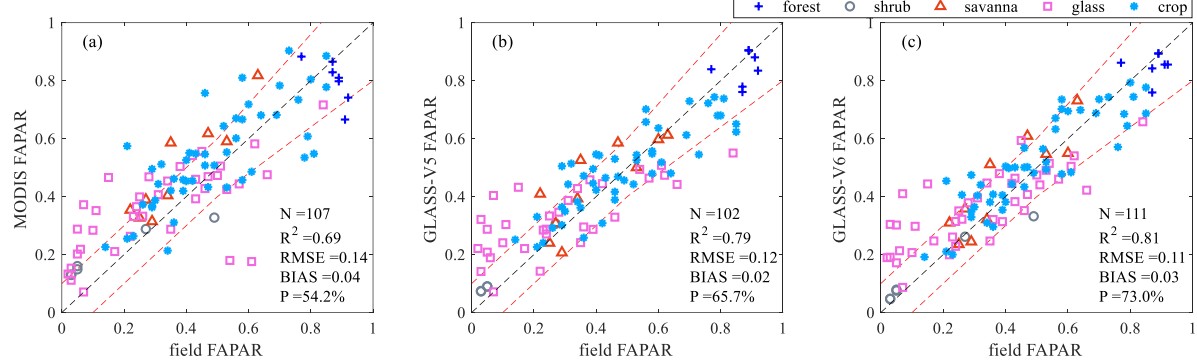

**Figure 6:** Direct validation of the (a) MODIS, (b) GLASS V5, and (c) GLASS V6 FAPAR products at the 3-km scale using the DIRECT ground measurement dataset from 2000 to 2017

**Table 2:** Direct validation of the four FAPAR products at the 250-m, 500-m, and 3-km scales using FAPAR reference maps from 2000 to 2016 for different biome types

| | | PROBA-V 250 m | *GLASS V6 250m | GLASS V6 250m | MODIS 500 m | GLASS V5 500m | GLASS V6 500m | MODIS 3 km | GLASS V5 3 km | GLASS V6 3 km |
|---|---|---|---|---|---|---|---|---|---|---|
| forest | $R^2$ | 0.76 | 0.90 | 0.78 | 0.55 | 0.59 | 0.73 | 0.54 | 0.05 | 0.01 |
| | **RMSE** | **0.09** | **0.09** | **0.09** | **0.12** | **0.11** | **0.09** | **0.13** | **0.07** | **0.06** |
| | Bias | -0.01 | 0.06 | 0.02 | 0.07 | 0.05 | 0.04 | -0.08 | -0.03 | -0.02 |
| | N | 415 | 415 | 1858 | 410 | 432 | 432 | 7 | 7 | 7 |
| | P | 83.9% | 73.3% | 73.8% | 63.9% | 57.6% | 75.9% | 85.7% | 100.0% | 100.0% |
| shrub & savanna | $R^2$ | 0.49 | 0.20 | 0.68 | 0.62 | 0.69 | 0.69 | 0.81 | 0.87 | 0.86 |
| | **RMSE** | **0.09** | **0.14** | **0.11** | **0.10** | **0.10** | **0.10** | **0.12** | **0.08** | **0.08** |
| | Bias | -0.03 | -0.08 | -0.04 | 0.01 | -0.01 | -0.04 | 0.09 | 0.04 | 0.02 |
| | N | 1137 | 1137 | 2249 | 560 | 601 | 601 | 15 | 15 | 16 |
| | P | 87.2% | 66.4% | 76.0% | 77.0% | 82.4% | 76.9% | 46.7% | 80.0% | 81.3% |
| grass | $R^2$ | 0.82 | 0.87 | 0.80 | 0.82 | 0.67 | 0.84 | 0.45 | 0.56 | 0.64 |
| | **RMSE** | **0.12** | **0.10** | **0.11** | **0.11** | **0.13** | **0.10** | **0.16** | **0.15** | **0.14** |
| | Bias | 0.03 | 0.01 | 0.00 | 0.05 | 0.00 | 0.01 | 0.05 | 0.04 | 0.04 |
| | N | 1381 | 1381 | 2065 | 529 | 546 | 546 | 37 | 32 | 40 |
| | P | 70.4% | 77.8% | 74.3% | 68.8% | 58.4% | 77.3% | 43.2% | 50.0% | 57.5% |
| crop | $R^2$ | 0.79 | 0.83 | 0.82 | 0.60 | 0.64 | 0.79 | 0.60 | 0.72 | 0.77 |
| | **RMSE** | **0.13** | **0.10** | **0.10** | **0.15** | **0.13** | **0.10** | **0.13** | **0.10** | **0.09** |
| | Bias | 0.08 | 0.06 | 0.04 | 0.08 | 0.06 | 0.04 | 0.05 | 0.00 | 0.02 |
| | N | 5858 | 5858 | 10891 | 2460 | 2678 | 2678 | 48 | 48 | 48 |
| | P | 59.6% | 69.2% | 70.6% | 47.7% | 56.6% | 69.7% | 60.4% | 66.7% | 79.2% |

 *GLASS V6 250 m: validation results using the same data points used for PROBA-V

Table 2 lists the accuracies of each FAPAR product for the different biome types at three scales validated using the reference data from 2000 and 2017, except for PROBA-V (2014–2016). The RMSEs of the GLASS V6 FAPAR are 0.06–0.14, and all of them are less than those of the GLASS V5, MODIS, and PROBA-V at the three scales for the different biome types, except

for the shrub and savanna biomes at the 250-m scale.

In terms of assessing the accuracy of the product in terms of meeting the target requirement, the GLASS-V6 FAPAR had the highest percentage of pixels with P values of greater than 70% at all three scales, while the PROBA-V, MODIS, and GLASS V5 FAPARs had lower percentages with P values of 54% to 66%.

### 4.3 Spatial consistency evaluation

Examples of the global spatial distributions of the MODIS, PROBA-V, GLASS-V5, and V6 FAPAR products in January and July of 2018 are shown in Fig. 7. The four products were found to have similar spatial consistencies. It should be noted that the grey areas correspond to missing data for the vegetated land surface. The MODIS and PROBA-V FAPARs contain missing data in January in the high latitude region of the Northern Hemisphere, which is due to cloud/snow contamination or the weak signals of the satellite observations. The GLASS V5 FAPAR cannot provide values above 70°, which is the same as the GLASS

V5 LAI, due to the poor representation in these areas during the machine learning model development. The GLASS V6 is the only product that provides full land coverage in winter and summer. The first reason for this is that the input GLASS V6 LAI is spatially and temporally continuous globally, and the second is that the Bi-LSTM model can extract the information from the entire time-series and assign a value to each time step provided that valid surface reflectance and LAI data are input.

The spatial distributions of the FAPAR products in two cloud-dominated and two middle-high-latitude regions are shown in Fig. 8. For the Yungui district in southern China, the MODIS and PROBA-V products had large missing data rates of 56% and 51%, respectively, while the GLASS products were spatially complete. In central Africa, the MODIS product had a missing rate of 15%, the PROBA-V product provided more valid data than the MODIS product, but it had lower values compared to the GLASS products. In central Europe, the GLASS V6 is more consistent with PROBA-V regarding textures and values. In

Alaska, MODIS contains 17% data gaps, and PROBA-V exhibits higher FAPAR than others in the lower right regions. By comparing the GLASS V5 and V6 products, it was found that V6 was spatially smoother than V5, and it exhibited more details such as the outline of the river. The reason GLASS V5 contains more noise is that it is based on the traditional surface reflectance filtering process, which inevitably introduces noise and uncertainties, especially in cloud-dominated areas. These cases further confirm that the GLASS V6 product had the highest spatial consistency among the current products in the cloud-

dominated areas.

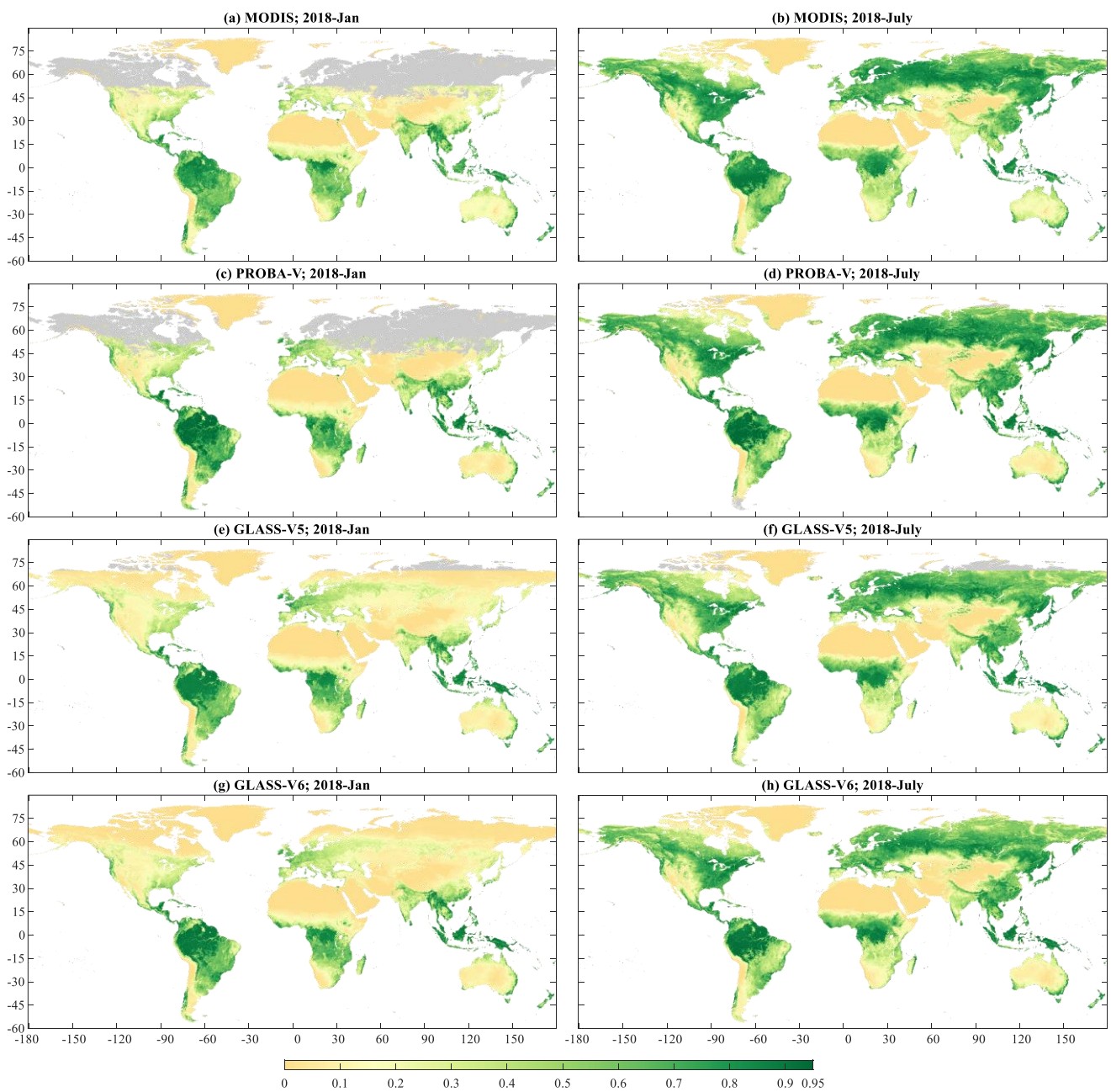

**Figure 7**: Global spatial distributions of the MODIS, PROB-V, GLASS V5, and V6 FAPARs in January and July of 2018. The spatial resolution is 0.05° latitude/longitude. (a) MODIS, January 2018; (b) MODIS, July 2018; (c) PROBA-V, January 2018; (d) PROBA-V, July 2018; (e) GLASS V5, January 2018; (f) GLASS V5, July 2018; (g) GLASS V6, January 2018; and (h) GLASS V6, July 2018


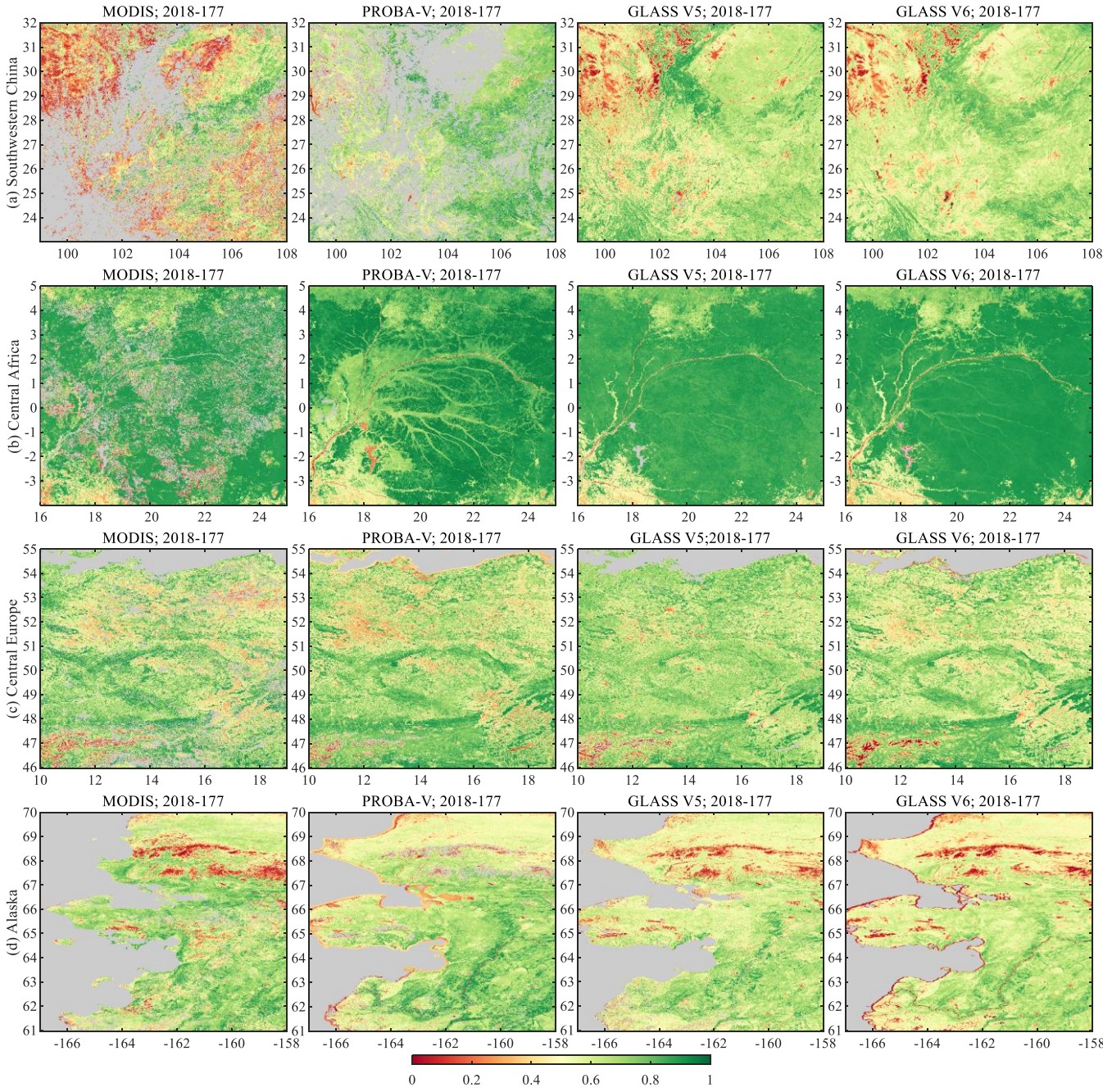

**Figure 8**: Spatial distribution of MODIS, PROB-V, GLASS V5, and V6 FAPARs on day 177 in 2018 in four regions: (a) Yungui district in Southwestern China; (b) Congo rainforest in central Africa); (c) Central Europe;and (d) Alaska. The spatial resolution is 500 m latitude/longitude

## 4.4 Temporal consistency evaluation

The time series of the MODIS, PROBA-V, GLASS V5, and V6 FAPAR products at eight DIRECT and Ameriflux sites are shown in Fig. 9. The spatial representativeness of the field FAPAR at the DIRECT sites (Figs. 9a–f) is 3 km, and therefore, the aggregated 3-km FAPAR products were plotted. Since the measurements from the Ameriflux sites (Figs. 9g–h) are at the tower footprint scale, the corresponding 500-m FAPAR products were extracted and plotted. During the growing seasons, the MODIS, PROBA-V, and GLASS products agree well with each other, but the PROBA-V FAPAR has lower values and the MODIS has discrete nonstable time-series profiles at the Counami site (Fig. 9b). At the Fundulea cropland site, which has multiple growing seasons, the MODIS, PROBA-V, and GLASS V6 products reflect the multiple growing seasons of the vegetation, while the GLASS V5 product is less sensitive to the seasonal variations. At the US-Ne2 cropland and US-Bar forest sites, for which continuous field FAPAR data are available, the four FAPAR products are closer to the field data at the US-Bar site than at the US-Ne2 site. This is due to the spatial representativeness of the tower measurements and the fact that the US-Bar site has a larger footprint and is more homogeneous than the US-Ne2 site (Tao et al., 2015).

The time-series FAPAR of the four products at their original spatial resolutions and the field references at 10 NEON sites are shown in Fig. 10. Generally, the MODIS and GLASS V6 products agree with the field references better than the PROBA-V and GLASS V5 products. At the NEON forest sites, the four products are slightly underestimated compared to the field values. The upper envelope of the MODIS product is closer to the field reference values, but it contains more noise than the other three products. The GLASS V6 FAPAR has more realistic seasonal trends than the V5 product, especially at the SCBI site (Fig. 10g), where the V5 FAPAR has abnormal seasonality. This is caused by the input of the GLASS V5 LAI data, which has been found to exhibit unrealistic seasonal variations in high latitude areas. Based on the analysis of the products at these typical sites, the GLASS V6 product was found to have more stable and continuous time-series trends than the other products, and it is an obvious improvement compared to the older version.

The GLASS V6 time-series FAPARs at one 1°×1° region in Southwestern China in 2021 are shown in Fig. 11, the seasonal variations of vegetation FAPAR can be clearly observed, consistent with the previous evaluation results, the GLASS V6 FAPAR product is spatiotemporally seamless at this cloud-dominated region.

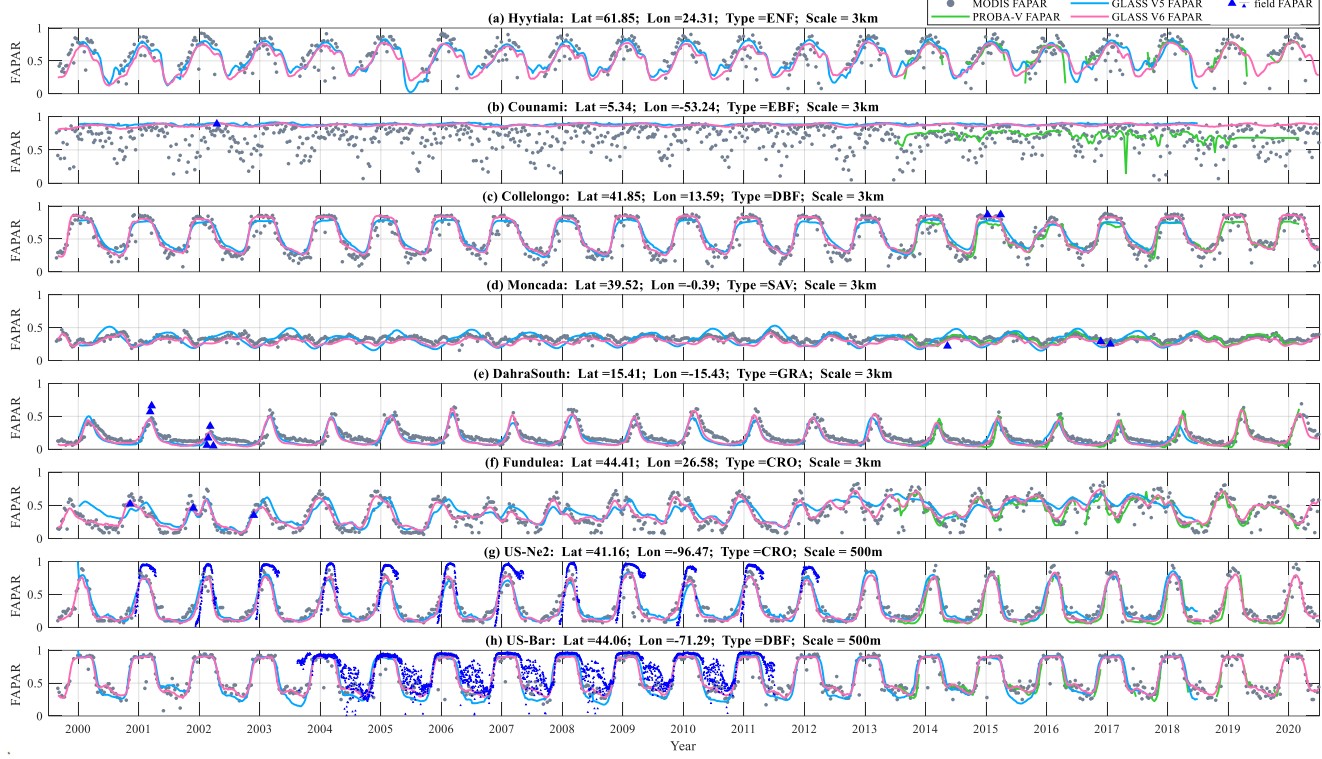

**Figure 9:** (a–f) Aggregated 3-km time-series MODIS, GLASS V5, GLASS V6, and PROBA-V FAPAR products at six DIRECT sites with different types of biomes during 2000–2020, and (g–h) 500-m time-series FAPAR products at two Ameriflux sites


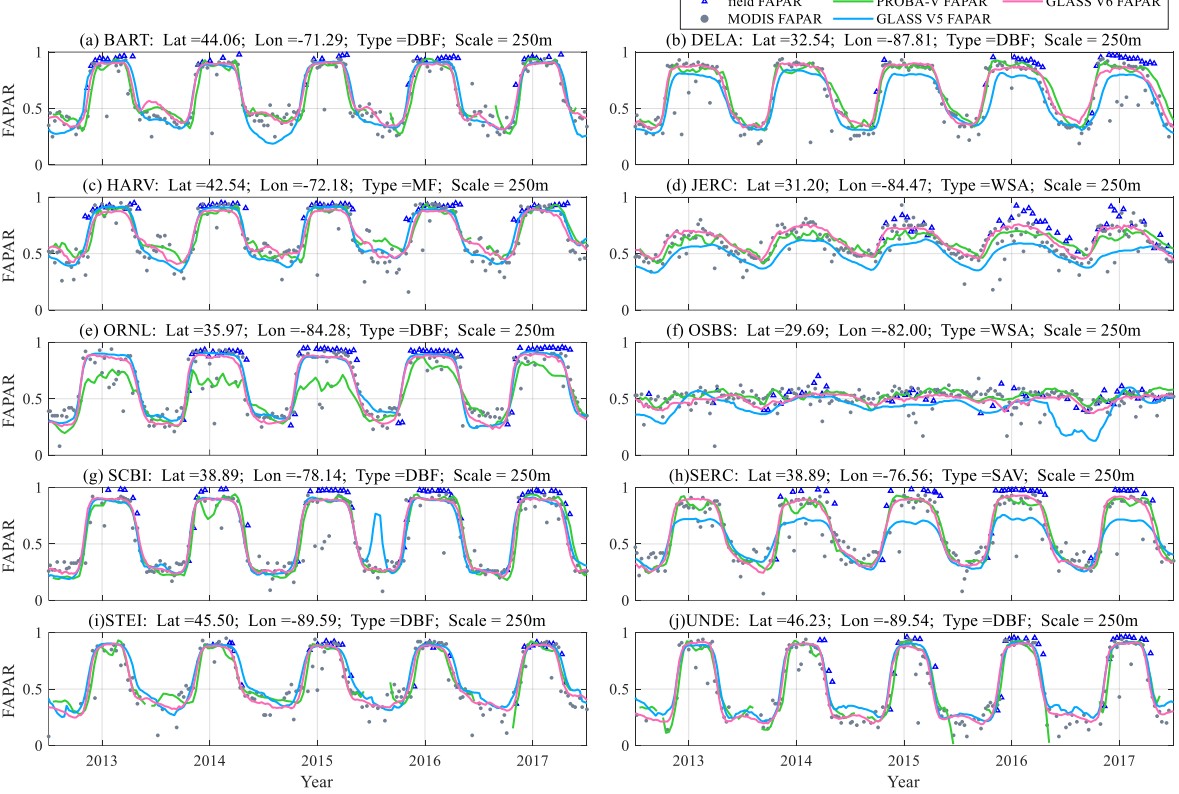

**Figure 10:** Time-series FAPARs from the MODIS 500-m, GLASS V5 500-m, GLASS V6 250-m, and PROBA-V 300-m products at 10 NEON sites during 2014–2018

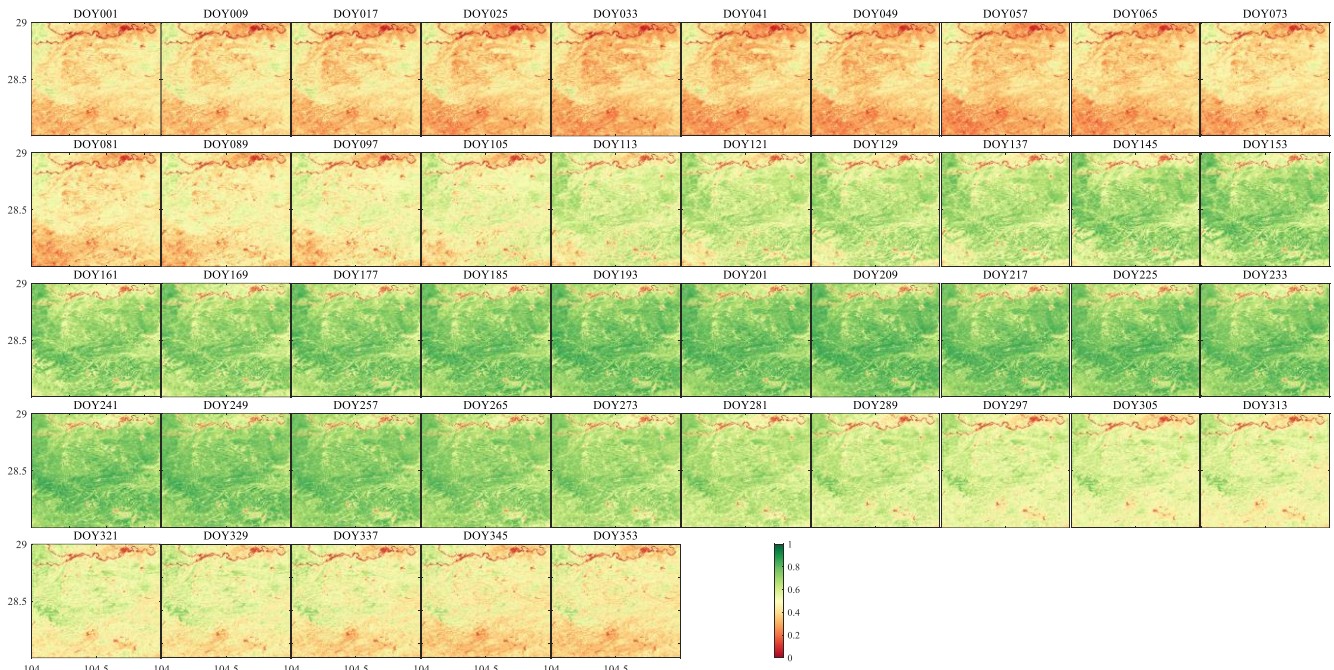

**Figure 11:** GLASS V6 time-series FAPARs at a 1°×1° region in Southwestern China in 2021, the spatial resolution is 250 m latitude/longitude

## 5 Data availability

The 250 m/8 day GLASS V6 FAPAR product for 2020 is freely available at https://doi.org/10.5281/zenodo.6405564 and https://doi.org/10.5281/zenodo.6430925 (Ma, 2022b, a), and at the University of Maryland for 2000-2021 (http://glass.umd.edu/FAPAR/MODIS/250m, last access 1 November 2022). We have also aggregated it to coarser resolutions (500 m/8 day, 0.05°/8 day, 0.1°/month, and 0.25°/month, http://glass.umd.edu/FAPAR/MODIS/, last access: 1 November 2022). The 250-m and 500-m data are in the sinusoidal projection, while the 0.05°, 0.1°, and 0.25° data are in the geographic

latitude and longitude coordinate system. The data files are provided in HDF-EOS format.

The GLASS V6 LAI dataset was downloaded from http://www.glass.umd.edu/ (last access: 1 November 2022). The MODIS and PROBA-V products were downloaded from https://earthdata.nasa.gov/ (last access: 1 November 2022) and https://land.copernicus.eu/global/products/fapar (last access: 1 November 2022), respectively.

## 6 Conclusions

In this study, the GLASS FAPAR product with a 250-m spatial resolution was derived from MODIS surface reflectance data. To our knowledge, this is the global long-time-serie FAPAR product for 2000–2021 with the highest spatial resolution. The time-series FAPAR samples used for the model training were created by merging the existing MODIS, PROBA-V, and GLASS

V5 FAPAR products using global distributed representative samples. The red and NIR bands, the solar zenith, view zenith, and relative azimuth angles obtained from the 250-m MODIS surface reflectance product, and the 250-m GLASS LAI data

were used as the features to predict the FAPAR values. The time-series GLASS V6 FAPAR was generated using the trained model at an optimal temporal length of 3 years. The accuracy and spatiotemporal consistency of the GLASS V6 FAPAR was quantified and evaluated through validation against field reference data and was compared with the MODIS, PROBA-V, and GLASS V5 FAPAR products. Through validation using 62 high-resolution FAPAR reference maps from the VALERI and ImagineS networks and 111 reference values from DIRECT, the GLASS V6 FAPAR product was demonstrated to have the

best agreement with the reference data, with an $R^2$ value of 0.80 and RMSEs of 0.10–0.11 at the 250-m, 500-m, and 3-km scales, and the highest percentage (72%) of retrievals in terms of meeting the accuracy requirements. In terms of the spatiotemporal consistency evaluation, the GLASS V6 FAPAR was demonstrated to have global coverage without data missing, maintain high quality even in cloud-dominated areas, exhibit consistent temporal profiles, and to reflect the seasonal variations of the vegetation well.


The GLASS-V6 FAPAR product is the first global long-term FAPAR product with a resolution of 250 m. The higher quality of the GLASS V6 FAPAR is attributed to the ability of the Bi-LSTM model to exploit the advantages of the existing FAPAR products, as well as to extract the temporal and spectral information from the MODIS observations and the GLASS LAI product. However, the accuracy still needs to be improved to fully meet the GCOS accuracy requirements. One source of

uncertainty is related to the differences between the FAPAR definitions of the three FAPAR datasets used in the model training process. These differences may lead to a slight uncertainty, which was not explicitly accounted for. More ground measurements for different ecosystems under various conditions are needed to further evaluate the FAPAR product.

## 7 Author contributions

HM performed data curation, investigation, methodology, validation, and writing-original draft; SL performed

conceptualization, supervision, writing- review and editing; CX and BL participated in data resources and validation; QW and AJ participated in data production and distribution.

## 8 Competing interests

The authors declare that they have no conflict of interest.

## 9 Acknowledgments

This study was partially supported by the National Key Research and Development Program of China (No. 2016YFA0600103) and China Postdoctoral Science Foundation Grant (2019M652707). We would like to thank the principal investigators of VALERI, ImagineS, and NEON networks for providing FAPAR reference data.

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
