# Peer review of "Global land surface 250-m 8-day Fraction of Absorbed Photosynthetically Active Radiation (FAPAR) product from 2000 to 2021"

_Earth System Science Data, 2022_

## Author Comment (AC1)

The authors develop a deep learning model to generate the new version of global 250 m spatio-temporal continuous GLASS FAPAR product. This product has great application potential as it is the first 250-m product and has the highest validation accuracy among existing FAPAR products according the evaluation results.

The paper gives details of the model, data, and product validation. Although the authors have done a good job, some modifications are needed to further their manuscript. Please see below for my specific comments:

Response: Thank you for your comments and suggestions. We have made revisions accordingly.

1. Reference for Ameriflux should be given.

Response: Added, thanks (see Line 160).

2. L222: In 2000, MODIS reflectance has no data before DOY49, did it affect the estimates in 2000? If added other 500-m bands, how the model accuracy changes?

Response: Thanks for raising these important questions. For the first concern, as MODIS began collecting data on February 24, 2000, for the calculation window of years 2000-2002, the missing data at the beginning days of 2000 was substituted by 2001 to fit the trained model, and the final FAPAR product begins from DOY 57. We have added this information in the text (line 240).

For the second, as we pointed out earlier, we used the 250-m surface reflectance to produce the 250-m FAPAR data, however, it only provides the red and NIR bands, therefore, only the first two red and NIR bands, solar zenith, view zenith and relative azimuth angles of the 500m MODIS surface reflectance product were used to train the model. If the other 500-m bands were added, we cannot apply the trained model for 250-m FAPAR production.

3. In Fig. 4, the meaning of P values should be explained.

Response: Thanks for this reminder, the metric P is used to represent the percentage of pixels meeting the target accuracy requirement in our validation, and we have added it in the text (line 275).

4. In Fig.6, "Direct validation of (the a) MODIS" should be "Direct validation of the a) MODIS".

Response: Thanks and revised.

5. How does the author aggregate the 250-m FAPAR to 500-m and 3-km? If using all MODIS surface reflectance bands to estimate 500-m FAPAR, can the accuracy be improved?

Response: Thanks, we aggregated the 250-m FAPAR to resolutions of 500 m and 3 km by averaging. (Added in line 245). Actually, the seven-band model for estimating the 500-m

FAPAR has a similar accuracy to the two-band model combined with GLASS LAI as another feature dimension, yet the seven-band model has low computational efficiency as it requires 4 more feature bands than our two-band model. Because we focused on estimating 250m FAPAR, we did not incorporate this result in the manuscript.

6. From Fig. 9, the GLASS V6 FAPAR was smoothed, did the author reprocess the estimated FAPAR curve, or the training samples are inherently smooth? How to reprocess them?

Response: Thanks. We did not perform other smooth processing on the estimated FAPAR. At the global representative pixels, the FAPAR training samples were created from MODIS, GLASS V5 and PROBA-V, as GLASS V5 and PROBA-V have relatively smooth temporal profiles because of pre- or post- process, the created FAPAR samples inherited both their smoothness and accuracy at the selected sample pixels. We have added more information on this point in the manuscript (line 190).

7. In Fig. 7, please check the number of the sub figures. The range of legend should be 0-1.

Response: Thanks and we have revised the figure captions. Actually, at the $0.05°$ spatial resolution, the maximum of the FAPAR is not larger than 0.95 according to our statistics. Therefore, we set the range of the color bar in Fig. 7 from 0 to 0.95, but in Fig.8 which displays the 500-m resolution FAPAR maps, the range of the color bar is from 0 to 1.

8. In Fig. 9, note the legend of field FAPAR (two triangles)

Response: Thanks, we display the field FAPAR in two sizes of blue triangles, as the number of field FAPAR in Fig.9g-h is much larger than in Fig.9a-f.

9. Do these four FAPAR products have the same definitions?

Response: Thanks for raising this important question. As we pointed it out in section 2.1, MODIS FAPAR corresponds to the instantaneous black sky FAPAR values (i.e., under direct illumination) at 10:30 solar time, PROBA-V FAPAR is defined as the instantaneous black sky FAPAR values (i.e., under direct illumination) at 10:00 solar time, and GLASS V5 FAPAR represents the clear-sky FAPAR at 10:30 a.m. local time. Although the FAPAR definitions of MODIS, PROBA-V and GLASS V5 products are somewhat different, according to previous studies, the impact of these differences on FAPAR's largest difference is less than a few percent compared to the uncertainties of the products (Martínez et al., 2013; Weiss et al., 2007). Therefore, the differences in the FAPAR definitions were ignored in this study. The GLASS V6 FAPAR is defined as black-sky FAPAR around 10:30 a.m. local time, which is an approximation of the daily average FAPAR.

10. L210: What is the resolution of input GLASS V6 LAI and output FAPAR? The input MODIS surface reflectance is 250 m, the output FAPAR should be also 250 m. How to get the 250-m FAPAR samples?

Response: We used the 500m GLASS LAI to train the model, as the created FAPAR "true values" samples are at 500m resolution. But in FAPAR production, we used the 250-m MODIS reflectance and 250m GLASS LAI as inputs, so the output FAPAR is at 250-m spatial resolution.

11. The GLASS V6 LAI product has uncertainty, the error will be brought into the estimation model. If only MODIS reflectance is used for training data, this error can be ignored, the accuracy of fAPAR maybe improved.

Response: Thanks. The model evaluation results in 4.1 indicate that incorporating GLASS V6 LAI as one feature of the input improves the accuracy of the model. This finding is also consistent with those of many previous studies, that is, the LAI is an important variable for estimating the FAPAR. Also, by incorporating GLASS LAI, the GLASS FAPAR product will keep consistent with LAI, which is important as discrepancies in these two important vegetation biophysical products may lead to errors in further applications.

---

## Author Comment (AC2)

The authors presented a method to generate a global FAPAR product at 250 m by exploiting MODIS surface reflectance data and deep learning model. The generated product is extremely interesting and useful. The validation analysis demonstrates the quality of the obtained product.

However, I have some comments to make to the authors:

1.  The reasoning for choosing the bi-LSTM method is not clear. Currently, deep learning models with transformer have outperformed LSTMs when working with both short and long time series of data.

Response: Thanks very much for your comments and suggestions. In our previous work in generating the GLASS new version LAI, we have demonstrated that the Bi-LSTM model outperforms the general regression neural network (GRNN), LSTM, gated recurrent unit (GRU) in learning the temporal relationship between satellite surface reflectance and vegetation variable. We agree transformer with attention mechanism is a more state-of-the-art deep learning system than LSTM and GRU, however, to keep consistent with the GLASS LAI product, we applied the same strategy and deep learning model in producing the GLASS 250-m FAPAR product from MODIS data. We have added more information on this in Introduction (line 85).

2.  It would be interesting to have more images like Figure 8 to really see (also qualitatively) the product obtained compared to the existing one.

Response: We appreciated this suggestion and added two more regions for intercomparison in Fig. 8.

3.  The authors stated that the obtained product has better spatiotemporal consistency. It would be nice to have figures showing how the FAPAR changes spatially for sequential instances of the time series (e.g., to see the FAPAR generated at time t1, t2, t3 and so on).

Response: Thanks for this valuable suggestion. We have added GLASS V6 250-m time-series FAPARs at one 1°×1° region in Southwest China in 2021 in Fig. 11 to show its spatiotemporal consistency.

4. What is the computational burden required to generate a new FAPAR product on a global scale once new MODIS surface reflectance data are available?

Response: We will update the GLASS FAPAR products yearly. As we stated in section 3.3, it takes about 48 h for calculating global tiles of three years using a single GPU. When new MODIS reflectance data are available, we will first update the GLASS 250m LAI, then GLASS FAPAR data. Therefore, the computational efficiency of Bi-LSTM is quite an advantage compared to the traditional algorithms.